# Implementing Technologies: Assessment of Telemedicine Experiments in the Paris Region: Reasons for Success or Failure of the Evaluations and of the Deployment of the Projects

**DOI:** 10.3390/ijerph20043031

**Published:** 2023-02-09

**Authors:** Alicia Le Bras, Kevin Zarca, Maroua Mimouni, Isabelle Durand-Zaleski

**Affiliations:** 1Hôtel Dieu Hospital, URC Eco Ile-de-France (AP-HP), Unité de Recherche Clinique en Économie de la Santé, 1 Place du Parvis Notre Dame, 75004 Paris, France; 2Department of Public Health, Henri Mondor-Albert Chenevier Hospitals (AP-HP), 94000 Créteil, France; 3Faculty of Medicine, University Paris-Est, 75000 Créteil, France; 4CRESS UMR 1153, 75004 Paris, France

**Keywords:** health policy, telemedicine, evaluation

## Abstract

Background: Telemedicine is increasingly viewed as a tool to provide a wide range of health services. This article presents policy lessons drawn from the evaluation of telemedicine experiments conducted in the Paris region. Methods: We used a mixed method design to study telemedicine projects commissioned by the Paris Regional Health Agency between 2013 and 2017. We combined data analysis of the telemedicine projects, review of the protocols, and interviews with stakeholders. Results: We identified the following reasons for disappointing outcomes: the outcome measure was requested too early during the experiments because payers required information for budgetary decisions; and the learning curve, technical problems, diversion of use, insufficient number of inclusions, and a lack of adherence prevented the demonstration of successful outcomes of the projects. Conclusion: The evaluation of telemedicine should be undertaken after sufficient uptake to ensure barriers to implementation are overcome, and to obtain the sample size necessary for statistical power and reduce the average cost for one telemedicine request. Randomized controlled trials should be encouraged with appropriate funding and the follow-up period should be extended.

## 1. Introduction

Telemedicine involves the remote exchange of data between professionals (tele-expertise and tele-assistance) or between patients and health care professionals (telemonitoring and teleconsultation). It is a rapidly growing field in healthcare and a well-researched topic with dedicated journals and PubMed publications increasing from 500 in 2019 to 1500 two years later [1,2]. Studies have already shown that telemedicine can potentially reduce costs and travel times while preventing complications due to delayed medical interventions [3,4]. The opinion of health professionals is that telemedicine may fill a gap in health services and improve access to and timeliness of appropriate care [5]. Meta-analyses conducted by the Agency for Healthcare Research and Quality (AHRQ) found that telemedicine for remote monitoring, counselling, or communication in chronic conditions was associated with benefits in mortality and quality of life, as well as reductions in hospital admissions [6]. Another meta-analysis suggested that telehealth is not statistically significantly different to usual care in quantitative terms but has important benefits for patients’ peace of mind and confidence to appropriately access healthcare [7]. A cross-sectional analysis of randomized clinical trials in digital health found a 27% non-publication rate at 5 years and postulated this was due to either technology failure or negative results [8]. The SARS-CoV-2 pandemic has shown the potential of telemedicine solutions to ensure continuity of care. The digital transition of healthcare systems benefits in Europe from dedicated EU funding. Because of the large elderly population and the epidemic of chronic conditions, the objective of many experiments is to allow elderly persons to stay at home or to stay away from hospitals. Evidence from regions in the Netherland, Baltic countries, Scotland, Galicia, and Germany suggests that the integration and process of care are improved, with a mixed effect on the patients’ outcomes [9].

In France, telemedicine experiments involving elderly patients have mostly focused on patients at home, with the objective of ensuring that they are safety monitored and avoid emergent hospital admissions. The evidence from these projects is somewhat conflicting, with improvement in satisfaction and, possibly, but not systematically, reduction in hospital days. Another project aimed at improving the care pathways of geriatric patients in nursing homes is being implemented in the Rouen region [10]. 

The Paris Regional Health Agency (ARS under its French acronym) pilots and regulates the provision of healthcare and prevention throughout the capital region of France. It is the single authority in charge of healthcare policy in the capital region of France and covers a population of 12 million with a healthcare budget of EUR 30 billion. The region has 419 hospitals and 2000 long-term and social care institutions and a total of 190,000 healthcare professionals. Its missions cover prevention, healthcare delivery, and social care. A 5-year plan, with a total budget of EUR 14M was established to foster the development of telemedicine in two major directions: (1) promote the efficient utilization of health services in people with chronic illness and long-term conditions; and (2) improve access to specialized care in geographically or socially deprived populations. Populations selected on the basis of need for specialized medical advice or care and access problems were: neonates, nursing home residents, and, more generally, older people with long-term conditions and social care needs.

Fifteen telemedicine pilot projects were selected after a competitive bidding process and independently assessed. Out of those fifteen, five were not implemented as planned due to delays, changes in governance, or professional or technical difficulties, and the assessments of two others have already been published [11,12]. The protocols for the eight remaining projects have been described in another article; all were publicly funded [13]. Assessments were conducted two to three years after the initial deployment of telemedicine according to the published protocols.

The objective of this article is to analyze the reasons for failure and understand the policy implications of seven failed or partly failed telemedicine projects, providing additional evidence regarding the barriers to rigorous evaluation of e-health programs [14]. The focus of our research was, therefore, both the deployment of the projects and the implementation of the evaluations, to identify whether a failure to demonstrate an effect of telemedicine resulted from a failed evaluation or a failed project, and whether success usually meant both the project implementation and the evaluation worked.

## 2. Materials and Methods

The evaluation requested by the Ile de France regional authority concerned every funded project. We used a mixed method design, combining evaluation protocols and data analysis of the telemedicine (TLM) projects, review of the protocols, and interviews with stakeholders.

### 2.1. Quantitative Analysis: Interventions, Study Protocols, Endpoints

The quantitative evaluation concerned seven projects that were assessed by before-and-after quasi-experimental studies; four of the designs included a control group [15,16,17]. Each evaluation project had its own primary endpoints defined by the investigators. The characteristics of each project, trial registration numbers, and design are presented in Table 1. Multiple endpoints were defined based on the particular characteristics of each telemedicine project. Details of the effectiveness, safety, and cost calculations have been published [13]. In short, the operating costs were valued from the health provider perspective. These costs included the labor costs related to operation of the telemedicine intervention, professional and patient education and training, investment in equipment, and the cost of building alterations and the call center, where relevant. The IT companies were selected by a tendering process for each project and this resulted in heterogeneous infrastructure and operating costs. The main expenditure items of IT companies related to research and development, as no one had previously developed telemedicine software. As the number of projects increased, in order to reduce costs, the system was shared between five of the projects via a platform. The two projects that did not benefit from this platform were TLM-Pathology Expertise and TLM-Pathology Frozen sections. Endpoint selection used previous publications from the whole system demonstrator [18]. Endpoint analysis used segmented regression, an open-source routing machine, or statistical parametric and non-parametric tests to compare before and after periods [19,20].

Sample size calculations were used to demonstrate superiority or non-inferiority in the case of the two pathology projects. An ethical review was not considered necessary for French authorities for the following reasons: (1) because the data source was extracted from the population-wide health database with only anonymous data; and (2) the endpoints were the professional procedures and not the clinical outcomes.

### 2.2. Implementation of the Evaluation and Deployment of the Projects

We conducted interviews with the members in charge of supervising the implementation of TLM projects and with the principal investigators of each project recruited on a voluntary basis. We used the template of the developed questionnaire on professionals’ experience to identify relevant topics [17]. For interviews, we contacted all healthcare professionals (physicians, nurses, technicians) and administrators (hospital director) involved in each project. The following 10 questions were asked by face-to-face interviews or email:Have you experienced technical difficulties which may affect the quality of care delivered by the telemedicine service?Have you experienced difficulties in your collaboration with other professional groups in relation to the telemedicine service?Have you experienced difficulties in your collaboration with the staff at other institutions in relation to the telemedicine service?How would you describe the usability of the telemedicine application for you?Has the use of the telemedicine application had any effect on your use of time?Has the use of the telemedicine application had any effects on your tasks?Has the use of the telemedicine application had any effects on the communication within your institution?Has the use of the telemedicine application had effects on the communication with other institutions?Would you like to continue to use the telemedicine service?How would you describe your overall satisfaction with the use of the telemedicine service?

To analyze the difficulties encountered in the implementation of the evaluations, we used the framework provided by T Greenhalgh and J Russel and examined the baseline hypotheses about the effects that underlie the sample size calculation, the interacting variables, the study design, and the political environment [14]. The baseline hypotheses on effect size for each project were proposed by the health professionals, usually from their own experience; as the supporting literature was considered at the time insufficient or irrelevant, the hypotheses were re-examined with hindsight at the end of the process. The interacting variables were identified as system-level and patient-level. The reasons underlying the choice of the study design were analyzed from the perspectives of the investigators and the sponsor. Finally, in order to identify political agendas, we examined the projects’ leadership and professional involvement.

## 3. Results

LM-PROMETTED was excluded due to major delays in the deployment of the teleconsultation platform between the institutions, which did not allow the project to start early enough for the assessment to be performed. We therefore present the results of six experiments.

### 3.1. Quantitative Analysis

All projects, with the exception of TLM-Pathology Expertise, underwent a full evaluation according to the registered protocols. With the exception of TLM-Pathology Expertise, the telemedicine projects failed to show significant improvement in the primary endpoints selected, both clinical and economic. Results by project are presented in Table 2 and Figure 1. In the telepathology expertise project, the response time was dramatically shortened as image transfer avoided handling, postage, and transportation. In the frozen sections tele-expertise, the percentage of results provided within the 30 min threshold decreased because of the time necessary to upload the images on the server; the mean time was 24 mn before and 34 mn after.

### 3.2. Implementation of the Evaluation and Deployment of the Projects

The baseline hypotheses on effect size for each project were usually too optimistic, which led to insufficient sample size and power. The interacting system-level variables included delays in information transfer and the decision process, lack of coordination between stakeholder, and multiple co-morbid conditions at the patient level. The choice of the study design was constrained by the practical difficulties of a randomized design (even cluster randomization was not feasible because institutions in the intervention group were selected by the sponsor), budgetary limitations, and the time frame, which was pre-set at 3 years by the accounting period.

To assess the professional and political issues in the deployment of the projects, we contacted all stakeholders identified and seventy-one stakeholders, mainly doctors in addition to a few nurses and laboratory technicians, were interviewed orally or by email. In all projects, health professionals reported their satisfaction and wished to continue using telemedicine, which had become an essential tool in their management of patients. They also approved the transfer of skills and improved communication between professionals. However, several problems associated with telemedicine were identified. The routine use of telemedicine had been hindered by technical incidents and performance issues, such as connection and network problems as well as IT malfunctions linked either to the platform or to the local network. These frequent incidents limited the adherence of professionals. In addition, there was a high turnover of professionals in the departments concerned as the projects extended over a period longer than initially planned. After each departure, it was necessary to train a new professional who could be less motivated to participate in the TLM project than those involved in the start-up operation. In the context of tele-expertise, telemedicine often generated a significant work overload for the remote specialists who analyzed complex and time-consuming patient records, and consequently additional human resources were requested.

## 4. Discussion

With the exception of one project, all evaluations were conducted as planned within the time frame. This in itself can be viewed as a success since the design of the evaluations required the sponsor, the health professionals, TLM providers, and methodologists, and discussions usually extended over a one-year period.

All projects described in this article, with the exception of TLM-Pathology Expertise, did not show a significant impact of telemedicine on care consumption or diagnosis delay. The use of telemedicine consistently increased the cost per patient, while the benefits to patients or to the general organization of care were generally unproven. Our analyses of these experiments concern the implementation of both the experiments themselves and the evaluation protocols.

### 4.1. Concerning the Implementation of TLM

The stakeholder’s interviews and cost analyses highlighted the following:(a)Technical difficulties in the deployment of TLM

Technical difficulties limited the use of telemedicine by postponing its routine use and undermining long-term confidence in this technology among the team; this, in turn, led to performance of a lower-than-expected number of procedures and increased their unit cost. The volume of telemedicine procedures would have been higher and the associated human time lower if the implementation and debugging periods had been extended before the assessment was performed. In several projects, the high cost per patient was due to the purchase of telemedicine-specific equipment. This equipment will be increasingly shared for other uses since TLM is at the heart of the evolution of the practice of medical specialties. The average costs per patient presented in Table 2 are therefore now overestimated.
(b)Stakeholder involvement

The motivation and involvement of medical and nursing staff were limited by a high turnover in the medical departments and the program sponsor. Low use by professionals was often associated with inconsistent governance, conflicting interests of the participants, and lack of accountability.
(c)The deployment of telemedicine

In several projects, professionals highlighted the significant work overload that telemedicine generated. An increase in funding to pay for overtime and additional human resources could incentivize professionals and therefore potentially allow a faster routine use of telemedicine. The economic model ensuring the continuity of the system should be specified at the start of the project and shared with the professionals.

### 4.2. Concerning the Research Protocols

It is true that we largely validated the predictions made by T Greenhalgh et al., which is even more unfortunate because we had prior knowledge of this research [14]. The discussion process with the regional authority, TLM providers, and stakeholders resulted in compromises, which were unsatisfactory from a scientific viewpoint. It may be, however, that the process of negotiation over the protocols, and not the protocols themselves, was the outcome sought by the policy makers, in which case arguments about study designs and endpoints are less relevant.
(d)Evidentiary requirements for resource use

The underlying hypotheses for TLM projects were that they could reduce transportation costs and service use of the emergency department, whilst improving the process of care and care pathways. We found that these expectations were often not met due to diversion of use, selection and attrition bias, and underpowered experiments. Based on these studies, it would seem that teleconsultations were not a substitute but a complement to face-to-face consultations. Associated with an additional cost, the benefits obtained were totally different in nature from those expected. Moreover, the inability to randomize patients in our studies, while it should have resulted in fewer constraints for investigators, threatened the comparability of populations, in particular when the sample size was small. In addition, evidence regarding costs had poor external validity due to heterogeneous infrastructures and operating costs.
(e)Evidence for policy decisions

Providing evidence on the benefits of telemedicine can prove a challenge for several reasons. In France, health authorities require an evaluation of costs (medical and nonmedical) and medical effectiveness (e.g., mortality, quality of life) and favor innovations with equivalent costs for greater effectiveness or an equivalent effectiveness for lower costs. However, telemedicine impacts several dimensions (e.g., quality of care, accessibility, professional practice) [5,13], making it difficult to choose a single criterion to measure. There is a disconnect between existing evaluation models for telemedicine and the requirements of health authorities. In these projects, it would have been necessary to define a sufficiently large range of criteria to account for all dimensions of telemedicine, as suggested by K. Kidholm et al. [17]. Multi-Criteria Decision Analysis (MCDA) approach could be used but, to our knowledge, has not yet been used to assess telemedicine.
(f)Dealing with time constraints

In all seven projects, the potentially beneficial effects of preventive teleconsultation, such as rapid clinical decision making to reduce emergency use and unplanned hospitalizations, could not be measured over a time horizon of one year. It may be that significant effects could be identified over a longer period of use. Since financing was limited in time, teams were not confident that they could continue using telemedicine after the experiment, and this may have limited their motivation for the implementation of TLM. Telemedicine is a disruptive innovation that requires time to be appropriated by professionals. Since the assessment of the projects was requested after a short period, the use of telemedicine was not yet fully integrated into clinical care and there was not enough time to resolve all of the early technical problems. The consequences of this short time horizon were insufficient number of inclusions and a possible lack of adherence. The time constraints led to compromises in the statistical analyses. For some studies, we could not include enough patients to reach the planned sample size, resulting in insufficient power and a limitation on the appropriate statistical models, particularly in terms of adjustment. The identification of significant effects requires data collection over a longer period. The time constraints did not allow conducting pilot studies, as is recommended for complex interventions and as is particularly relevant for digital health when participation, delivery, and retention of the intervention are of particular concern [21,22]. 

## 5. Recommendations

Because telemedicine modifies the workflow and interactions between professionals, support should be provided during its deployment until it is well accepted and seen as a facilitator of, rather than an impediment to, seamless care pathways.

Negative results or unpublished results are not uncommon in digital health interventions. Failure in evaluations of e-health programs had been reported even before the Ile de France experiments begun, due to the multiplicity of goals and stakeholders, the instability of the intervention and its outcomes over time, the decision-making process, which requires measurable quantitative and cost outcomes, and the constrained time frame because budgets need to be approved yearly [14]. Our conclusions for a narrower group of complex interventions than-e health as a whole lead us to recommend different approaches to the assessment of TLM projects.

TLM projects that aim to improve morbidity or mortality endpoints should probably not be conducted at the regional level but at the national level to ensure a significant budget is at the project’s disposal and a large number of patients can be included [23]. In contrast, feasibility endpoints may be sufficient for small-scale projects that aim to better coordinate care among healthcare institutions.

The research protocols used to assess the effectiveness and efficiency of the projects must be in line with the research literacy of the participating centers. Randomized trials may not be appropriate in institutions unfamiliar with research. To identify an adequate primary endpoint, it is necessary for health authorities to work hand-in-hand with users and researchers in order to find a compromise between the best measure of the expected benefit and what can realistically be collected.

## 6. Conclusions

In considering the effectiveness and cost effectiveness of complex interventions such as telemedicine or e-health, it might be useful to consider that the journey is the destination; in other words, the process of discussing evidence development, study design, and endpoints is part of the objective of the policy makers to spread the culture of health technology assessment amongst all categories of health and social care professionals. Some TLM applications that involve healthcare professionals already familiar with research protocols may be more ready for full assessment than applications developed in other areas. It can be expected, however, that the deployment of TLM with the associated requirements for evaluation will speed up the uptake of health technology assessment in all areas of health, including social care.

## Figures and Tables

**Figure 1 ijerph-20-03031-f001:**
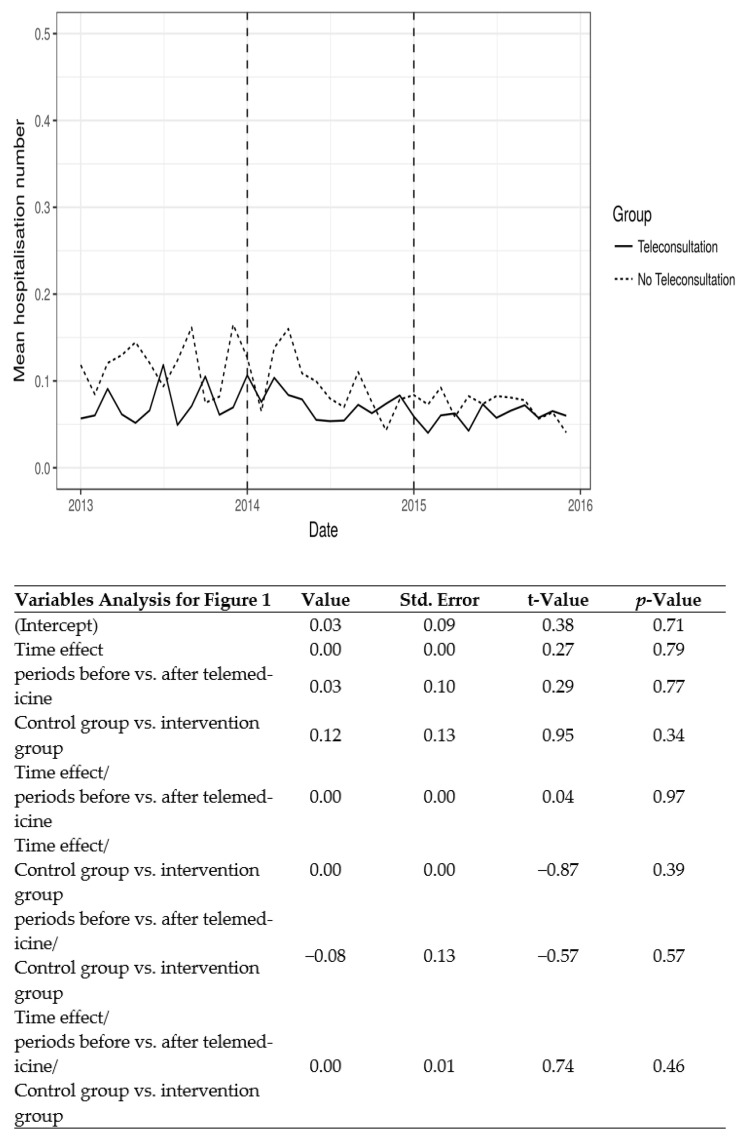
Mean unplanned hospital admissions in nursing home residents (projects TLM-TMG91 and TLM-E-VLINE) before and after teleconsultation became available (2014 for TMG91 and 2015 for E-VLINE).

**Table 1 ijerph-20-03031-t001:** Population, conditions and setting, rationale, design, intervention, comparator, and outcomes for telemedicine use.

Project and Objectives	Population and Trial Registration	Design	Intervention	Comparator	Outcomes
Teleconsultations: The elderly population is often exposed to unnecessary hospitalizations, which can cause deterioration in their health status.	Dependent polymorbid nursing home residents (TLM-TMG91- NCT02164747 and TLM-E-VLINE-NCT02157740) without on-site access to primary or secondary care. TMG: TeleMedicine Geriatrics; 91 is the department code E-VELINE is a play on words with the department’s name, Yvelines	Controlled before after. We compared two periods with a segmented regression analysis: before telemedicine (January 2013–December 2013 for TLM-TMG91 and January 2013–December 2014 for TLM-E-VLINE) and during routine use of telemedicine period (January 2015–December 2015 for TMG91 and January 2016-December 2016 for TLM-E-VLINE).	Programmed teleconsultations, either open to all medical specialties or with a psychiatrist or emergency teleconsultation with a doctor on call. 20 nursing homes.	Same number of nursing homes without telemedicine matched with propensity scores computed from variables identified by the Regional Health Agency and geriatricians: the number of private and public hospitals located within 20 min from the nursing home—computed using Open Source Routing Machine (OSRM)—the proportion of residents over 90 years old, the average level of dependence of residents, a global indicator of health care, the mean number of transportations and consultations.	Number of unplanned hospitalizations by nursing home and by month. Data was extracted from the national claims database, and aggregated at the nursing home level
Teleconsultations: Because of their severe disabilities and medical shortage, access to specialized consultations and preventive follow-up is limited.	Autistic children and adolescents (TLM-PROMETTED-NCT02996708) PROMETTED: (PROgramme MEdical de Télépsychiatrie)	Controlled before after (January 2014–December 2014) and after during routine use of TLM (January 2017–December 2017) comparison.	Programmed teleconsultation with a pediatric neurologist or a psychiatrist in five institutions benefiting from telemedicine	Three structures with no access to telemedicine	Proportion of patients who had taken the Autism Diagnostic Interview-Revised (ADI-R), a structured interview and rating score, at least once during the study, and at least one reassessment per year of the Childhood Autism Rating *Scale* (CARS) which rates items indicative of autism spectrum disorder after direct observation, and Vineland Adaptive Behavior scale, a measure of adaptive behavior skills for children and adolescents.
	children and adolescents with multiple handicaps (TLM-POLYHANDICAP) living in institutions	Before after in eight institutions treating children and adolescents with multiple handicaps. before telemedicine (January 2015–December 2015) and during routine use of TLM period (January 2017–December 2017).	In these institutions, the patients could have access to teleconsultation and/or consultation. All those who received at least one teleconsultation were included in the intervention group	The control group consisted of patients who had only received consultations, without benefiting from telemedicine.	The primary endpoint was the average number of neuropediatric visits per child.
Tele-expertise. Neonatology services need access to expert professionals in brain imaging, whether in the context of ongoing care or when a second opinion is required. Some decisions in neonatal resuscitation cannot be made without an expert *in* ethics *and* medico-legal *aspects*.	Newborns hospitalized in a neonatal intensive care unit with severe brain disorders (TLM-MATRIX NEONAT). MATIX: MAgneTic Resonance teleXpertise	Before after. Before telemedicine (December 2014–September 2015) and during routine use of telemedicine (June 2016–December 2016).	MRI image transfer for a second opinion from a pediatric neuroradiologist. The requests for expertise from six neonatal intensive care units were analyzed.	Before telemedicine	The primary endpoint was the time between the date of MRI and the date of decision; the decision could be a withdrawal of resuscitation, continuation of care, or the request for another MRI.
Second opinion from a pathologist specialist	Frozen section and images (TLM-Pathology Expertise-NCT02374697)	Before after. Before telemedicine (September 2013–December 2013) and during routine use of telemedicine (January 2015–June 2015).	Transfer of digital slides uploaded to a webserver for a second opinion from specialized pathologists for complex pathological diagnoses. The requests for expertise from 16 pathology units were analyzed.	Before telemedicine	The primary endpoint was the average response time to receive the results, i.e., the time between the dispatch of the digital slides to the reception of the report giving the second opinion of the remote pathologist,
*Medical tele-assistance* for intraoperative consultations during a surgical procedure; in some community hospitals, no on-site pathologist is available. Offsite analysis of images of intraoperative frozen sections had the goal of obtaining the same diagnostic accuracy as the original glass slide interpretation. The objective was to verify that the implementation of telemedicine did not extend delays for the result from the expected time of 30 min.	(TLM-Pathology Frozen Section –NCT02368769).	Before after. Before telemedicine (January 2013–June 2013) and during routine use of telemedicine (January 2015–June 2015).	Intraoperative frozen sections from a two-site academic department of pathology were analyzed.	Before telemedicine	The primary endpoint was the period of time between the specimen’s time of arrival to the pathological anatomy and cytology unit and the time of result’s transmission to the surgeon.

**Table 2 ijerph-20-03031-t002:** Results obtained in six telemedicine projects implemented by the Ile de France region. (TLM-PROMETTED was excluded).

Projects	Primary Endpoints	Design		With TLM			Without TLM		*p* Value
			N	Cost/Patient	Effectiveness	N	Cost/Patient	Effectiveness	
TLM-TMG91	Number of unplanned hospitalizations by nursing home and by month.	Controled before after	Control: 366	€1579	See Figure 1	Control: 292	331	See Figure 1	0.39
			Intervention: 397			Intervention: 341			
TLM-E-VLINE	Number of unplanned hospitalizations by nursing home and by month.	Controled before after	Control: 976	€1489	See Figure 1	Control: 1145	238	See Figure 1	0.45
			Intervention: 1087			Intervention: 1232			
TLM-POLYHANDICAP	Number of neuropediatric visits per child	Case control	31	984	1.3 (Before period)	27	372	1.3 (Before period)	0.99 (Before period)
					2.0 (After period)			2.0 (After period)	0.93 (After period)
TLM-MATRIX NEONAT	Time between the date of MRI and the date of decision	Case control	40	377	4.4 days	30	220	5.9 days	0.35
TLM-Pathology Expertise	Response time to the requesting pathologist	Before after	100	357	6.9 days	134	88	24.9 days	*p* < 0.001
TLM-Pathology Frozen Sections	Proportion of time less than 30 min between the time of arrival of the specimen in the pathological anatomy and cytology unit and the time of transmission of the result to the surgeon	Before after	98	181	0.48	89	148	0.81	0.33

## Data Availability

Data is available upon reasonable request to the corresponding author.

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
