# Peer review of "Implementing Technologies: Assessment of Telemedicine Experiments in the Paris Region: Reasons for Success or Failure of the Evaluations and of the Deployment of the Projects"

_ijerph, 2023, doi:10.3390/ijerph20043031_

Round 1
Reviewer 1 Report
General comments
There are a few minor things (shown in the detailed comments) that the authors could clean up, but overall, this is a good, thoughtful paper, despite the shortcomings of its sample and methods. I think that, despite some flaws, the paper tries, and succeeds, in contributing to the literature, and perhaps to the practice, of telemedicine.
Detailed comments
Line 117: This is the only place in the manuscript where ADI-R is mentioned. I think it would be helpful if you briefly explained what it is.
Lines 163-165: Where you have placed the legend for Figure 1 is a bit confusing, as it appears to be part of a paragraph in section 3.2. I would eliminate the 3.2 numbering here and put the Figure 1 legend under the figure itself.
Between lines 1665-167, the last two rows of Table 1: The final two rows of entries in Table 1 are oddly formatted. You might want to clean that up.
Am I reading Table 1 correctly that pathology expertise took 6.9 days before teleconsultation and 24.9 days after? This is statistically significant, but in a dramatically negative way, right?
And that the proportion of times below 30 minutes for frozen sections was 48% before TM and 81% after? It seems odd that such a big difference was not statistically significant, but I trust your calculations were correct. This result probably had something to do with small sample size, which you mention later in the manuscript.
Lines 202-204: Perhaps I am thinking of these wrongly, but could it not be said that the experiments were successful? In the sense that, except for Pathology Expertise, there was no statistically significant difference between in-person medicine and tele-medicine (at least on the measures you chose)?
Lines 247-249: Good point about the constraints of sample size. I thought of that as I read your results.
Author Response
There are a few minor things (shown in the detailed comments) that the authors could clean up, but overall, this is a good, thoughtful paper, despite the shortcomings of its sample and methods. I think that, despite some flaws, the paper tries, and succeeds, in contributing to the literature, and perhaps to the practice, of telemedicine.
We are very grateful to the reviewer for this comment.
Detailed comments
Line 117: This is the only place in the manuscript where ADI-R is mentioned. I think it would be helpful if you briefly explained what it is.
We apologize for this oversight. The explanations for the scales have been added and the paragraph now reads: The primary endpoint was the proportion of patients who had taken the Autism Diagnostic Interview-Revised (ADI-R), a structured interview and rating score for each question test at least once during the study and at least one reassessment per year of the Childhood Autism Rating Scale (CARS) which rates items indicative of autism spectrum disorder after direct observation and Vineland Adaptive Behavior scale is a commonly used measure of adaptive behavior skills for children and adolescents with a before telemedicine (January 2014 – December 2014) and after during routine use of TLM (January 2017 – December 2017) comparison.
Lines 163-165: Where you have placed the legend for Figure 1 is a bit confusing, as it appears to be part of a paragraph in section 3.2. I would eliminate the 3.2 numbering here and put the Figure 1 legend under the figure itself.
We agree with the reviewer and have reformatted this section, including a table as suggested by reviewer 3.
Between lines 1665-167, the last two rows of Table 1: The final two rows of entries in Table 1 are oddly formatted. You might want to clean that up.
Thank you
Am I reading Table 1 correctly that pathology expertise took 6.9 days before teleconsultation and 24.9 days after? This is statistically significant, but in a dramatically negative way, right?
This is the other way around, 6.9 days with telemedicine vs 24.9 days without.
And that the proportion of times below 30 minutes for frozen sections was 48% before TM and 81% after? It seems odd that such a big difference was not statistically significant, but I trust your calculations were correct. This result probably had something to do with small sample size, which you mention later in the manuscript.
That is correct, the increase in turnaround time was due to the time to upload the images. However the mean time only increased by 10mn, this was added in the Ms.
Lines 202-204: Perhaps I am thinking of these wrongly, but could it not be said that the experiments were successful? In the sense that, except for Pathology Expertise, there was no statistically significant difference between in-person medicine and tele-medicine (at least on the measures you chose)?
We agree with the reviewer that, form a political viewpoint and with the experience of the SARS-COV pandemic, telemedicine is being deployed based upon the overwhelming benefit to patients’ and professionals’ satisfaction.
Lines 247-249: Good point about the constraints of sample size. I thought of that as I read your results.
Thank you
Reviewer 2 Report
Dear Authors, the subject of the article is important and it is great that a good study is created. Therefore, I point out the following weaknesses of the text to improve:
1. I suggest changing the title to a more emphasizing research problem;
2. The introduction is too short. There is no proper review of the literature divided into world and French, plus who has already written about the same projects as you the authors chose for analysis;
3. The research method is basically not described. Although there is an explanation in subchapter 2.2, there is still no unification of the research concept;
4. No discussion of statistical analysis;
5. Tables and figures pasted as if by accident...;
6. The Discussion is loosely related to Results and Conclusions;
7. Incorrectly saved References.
I also include comments in the text.

Reviewer 3 Report
Thank you very much for the opportunity to review this article. I found it interesting and believe it performs an analysis infrequently in the scientific literature. However, some minor aspects are susceptible to revision. These are as follows:
In the introduction, the authors could describe the rationale for their work before describing the objectives: for example, because of the more than likely scarcity of similar work or the importance of drawing consequences that may help the development of similar projects.
Sections 2.1. and 2.2. on materials and methods are somewhat complicated to read. The readability of the text will be improved if the authors tabulate the information provided in these two sections.
In the methodology, it would be helpful if the authors detailed more the methodology followed for the analysis.
Despite the particular structure of the work developed, it would be helpful for the authors to explain the limitations and strengths of their work. As the authors point out, "TLM-PROMETTED was excluded due to major technical problems encountered that did not allow the assessment to be performed. We therefore present the results of six experiments". Therefore, this may be biasing the results and conclusions. Other potential limitations or biases should also be analyzed. This analysis would allow the authors to assess the robustness and external validity of the conclusions. This analysis would also enrich the discussion.
Author Response
Thank you very much for the opportunity to review this article. I found it interesting and believe it performs an analysis infrequently in the scientific literature.
We are grateful for this comment and thank the reviewer
However, some minor aspects are susceptible to revision. These are as follows:
In the introduction, the authors could describe the rationale for their work before describing the objectives: for example, because of the more than likely scarcity of similar work or the importance of drawing consequences that may help the development of similar projects.
We have added to the rationale of the work. There is indeed some literature on failed evaluations, most notably the publication by T Greenhalgh and J Russel, we attempted to understand the issues with both the evaluations and the projects themselves, in a slightly different context since we had first hand knowledge of the evaluation protocols and the difficulties encountered to have the protocols accepted by the stakeholders .
The focus of our research were therefore both the deployment of the projects and the implementation of the evaluations, as a failure to demonstrate an effect of telemedicine resulted from a failed evaluation or a failed project while a success meant both the project implementation and the evaluation worked .
Sections 2.1. and 2.2. on materials and methods are somewhat complicated to read. The readability of the text will be improved if the authors tabulate the information provided in these two sections.
We have tried to improve the readability with a table but are open to other presentations.

Round 2
Reviewer 2 Report
Dear Authors, the current title seems correct to me. The content after the revision also gained transparency. I suggest publishing.